# The Influence of Viral Infections on Iron Homeostasis and the Potential for Lactoferrin as a Therapeutic in the Age of the SARS-CoV-2 Pandemic

**DOI:** 10.3390/nu14153090

**Published:** 2022-07-27

**Authors:** Jeffrey L. Ward, Moises Torres-Gonzalez, Mary Cloud B. Ammons

**Affiliations:** 1Medical Student, College of Osteopathic Medicine, William Carey University, Hattiesburg, MI 39401, USA; jeffward.do@gmail.com; 2VP, Nutrition Research, National Dairy Council, Rosemont, IL 60018, USA; moises.torres-gonzalez@dairy.org; 3Associate Research Scientist, IVREF, Boise VA Medical Center, Boise, ID 83702, USA

**Keywords:** SARS-CoV-2, COVID-19, lactoferrin, anemia, inflammation, iron homeostasis

## Abstract

The association of hyperinflammation and hyperferritinemia with adverse outcomes in SARS-CoV-2-infected patients suggests an integral role for iron homeostasis in pathogenesis, a commonly described symptom of respiratory viral infections. This dysregulated iron homeostasis results in viral-induced lung injury, often lasting long after the acute viral infection; however, much remains to be understood mechanistically. Lactoferrin is a multipurpose glycoprotein with key immunomodulatory, antimicrobial, and antiviral functions, which can be found in various secreted fluids, but is most abundantly characterized in milk from all mammalian species. Lactoferrin is found at its highest concentrations in primate colostrum; however, the abundant availability of bovine-dairy-derived lactoferrin (bLf) has led to the use of bLf as a functional food. The recent research has demonstrated the potential value of bovine lactoferrin as a therapeutic adjuvant against SARS-CoV-2, and herein this research is reviewed and the potential mechanisms of therapeutic targeting are considered.

## 1. Introduction

Iron is vital to living cells because of the essential roles it plays in various biological systems such as cytochromes, oxygen binding molecules, and enzymes. Not only is iron necessary for proper cellular function, but its concentration must be carefully controlled, as both high and low iron levels can lead to cell injury and death [1,2]. The maintenance of iron homeostasis is primarily mediated through regulating dietary iron absorption by enterocytes and the release of recycled iron from macrophages. Iron homeostasis can be significantly impacted by infection and the host immune response. For example, severe systemic iron overload can result in an increased susceptibility to infection and the secretion of proinflammatory cytokines can disrupt iron homeostasis [3,4]. During infection and inflammation, including viral infections, macrophages become iron-overloaded [5]. To counter the viral disruption of iron homeostasis, the multifunctional glycoprotein lactoferrin has been proposed as a bioactive therapeutic, including most recently for the treatment of SARS-CoV-2 infections [6]. Herein, we review the dynamic interactions between viral infection, iron homeostasis, and inflammation, focusing specifically on the potential for lactoferrin as a therapeutic adjuvant for the treatment of COVID-19, the disease caused by infection with the SARS-CoV-2 virus.

## 2. Iron Homeostasis and Viral Infection

Dietary iron enters the body through enterocytes lining the intestinal wall via the enzymatic action of the duodenal cytochrome B, which catalyzes the reduction of Fe^3+^ to Fe^2+^, and the divalent metal transporter 1, which binds Fe^2+^. Ferrous iron is then released into the plasma through ferroportin [7]. Ferrous iron is converted to the ferric form by either the hephaestin or ceruloplasmin ferroxidases, found in the small intestine, or hepatocytes and macrophages, respectively, and bound to serum transferrin (Tf) for transportation and storage in cytosolic ferritin [8]. The regulation of iron homeostasis is dependent on the diet-mediating hormone hepcidin, which binds ferroportin, leading to internalization and degradation [9]. Increased iron in the serum leads to the increased expression of hepcidin, with concurrent retention of iron in macrophages and reduced absorption of dietary iron. Conversely, low levels of iron in the serum will trigger the suppression of hepcidin expression and increased release of intracellular iron [10]. Hepcidin is also an acute-phase protein, synthesized in response to inflammatory conditions through IL-6 [11], thereby linking iron homeostasis and inflammation through the regulation of hepcidin.

The mechanistic relationship between viral infection and iron homeostasis is not fully understood; however, viral infections can alter iron homeostasis (see Drakesmith and Prentice for an excellent review [12]). For example, inflammatory cytokines induced in viral infections can impact cellular and systemic iron concentrations by inducing hepcidin synthesis, blocking ferroportin assembly and transferrin release [13]. Influenza A, which uses the transferrin receptor protein (TfR1) to enter the cell, enhances the expression of the proinflammatory cytokine IL-6 (Figure 1) [14]. This increased production of IL-6 was reduced three-fold in the presence of the hydroxyl radical scavenger dimethylthiourea. During viral infections, particularly chronic viral hepatitis, circulating free iron has been detected [15]. Whether this circulating free iron is the result of hepatic cell death, the release of free iron from dying macrophages with iron overload, or a combination of both remains debatable. Because free iron can induce reactive oxygen species to further perpetuate the inflammatory response, this finding strengthens the idea that iron levels above the homeostatic parameters could potentially cause patients with viral infections to have less favorable outcomes due to an excessive inflammatory response [16].

While there are many different ways in which viruses can impact iron metabolism and alter iron homeostasis, the findings from human immunodeficiency virus (HIV-1) infection provide some insight. HIV-1 has evolved many mechanisms to support host cell iron retention to the benefit of long-term viral survival and replication. For example, the myristolated protein Nef of HIV-1 downregulates the macrophage major histocompatibility complex (MHC-1b) expression of the homeostatic iron regulator (HFE) protein. HFE is a membrane protein that helps regulates the circulating iron uptake by regulating the interaction of the transferrin receptor (TfR) with transferrin (Tf). Pathologically low levels of HFE result in ferritin iron accumulation in macrophages, which are inversely correlated with survival times [17].

Additional predictors of mortality in HIV-1 infection include genetic polymorphisms in iron regulatory genes, such as the *SLC11A1* gene, encoding the natural-resistance-associated macrophage protein-1 (Nramp1), and the *HP* gene, encoding the free-hemoglobin scavenger haptoglobin. Nramp1 is hypothesized to act as an iron gatekeeper in macrophages and is key to the host resistance to infection. *SLC11A1* gene mutations have been associated with increased susceptibility to infectious diseases, as well as diseases of chronic inflammation; however, the iron homeostasis contribution of Nramp1 and haptoglobin during HIV-1 infection remains an area of debate [18]. Finally, it has been demonstrated that the iron chelator deferoxamine decreases viral replication in HIV-1 infections by impacting the iron-dependent viral replication [19]. In addition to HIV, other viruses such as cytomegalovirus, members of the arenavirus family, and members of the picornaviridae family have been shown to be linked to disrupted iron homeostasis and dysregulated iron metabolism during infection [12,19].

## 3. Lactoferrin as a Therapeutic Adjuvant in Respiratory Viral Infections

### 3.1. Bovine Lactoferrin: A Multifunctional Glycoprotein

Increasingly, the bioactive value of dairy products has led to an expansion of dairy-derived functional foods on the market. Whey is derived from the cheese making process and used to be discarded as waste; however, the characterization of bioactive proteins in whey has revived interest in the market value of whey and its derivatives as nutraceuticals [20]. One such bioactive protein found in whey is the iron-binding glycoprotein lactoferrin. Lactoferrin (Lf) is produced by all mammals and can be found in most secretory fluids such as colostrum, milk, saliva, and tears [21], as well as neutrophil secretory granules [2]. At ~80,000 Da, lactoferrin is a larger member of the transferrin family of non-heme, iron-binding glycoproteins. With two structural lobe domains, Lf can bind two ferric ions (Fe^3+^) with high affinity within each lobular cleft (Kd = 10–20 M). Depending on whether Lf is iron-bound or unbound, the protein assumes either the open form (apo-lactoferrin) or closed form (halo-lactoferrin) [22]. The two lobes are connected by a short peptide, which forms a 3-turn α-helix that can be cleaved to produce the lactoferrin peptide derivatives lactoferampin and lactoferricin, which have been shown to have antimicrobial properties [22]. Originally thought to function as an iron transporter, the innate immune-modulating characteristics of Lf are increasingly appreciated, including inhibiting neutrophil priming by bacterial lipopolysaccharide (LPS), enhancing neutrophil adherence to endothelial cells, and modulating inflammation by amplifying apoptotic signals (referred to as ferroptosis) [23,24,25].

Dairy-derived bovine lactoferrin (bLf) is generally recognized as safe (GRAS) by the FDA and is commonly used as a nutraceutical dietary supplement [26]. Of the total milk proteins found in bovine milk, 20% are whey protein, with 1.5% of the whey protein consisting of bLf. In contrast, human milk consists of 60% whey proteins, with an average of 2.5% consisting of human lactoferrin (hLf) [27,28]. Despite the difference in lactoferrin abundance between humans and bovines, of all lactoferrin-producing non-primates, bLF is the most similar structurally to the lactoferrin that is endogenously produced by humans [21]. However, some distinctions between human lactoferrin (hLf) and bovine lactoferrin (bLf) have been demonstrated. For example, bLf is proposed to bind up to four times the amount of iron bound by hLf [29], and hLf contains multiple unique steroid response elements [30]. Additionally, differences in Lf distribution exist between species, with bLf being more abundant in saliva and produced by bronchopulmonary structures such as the serous and mucous cells of the bronchial glands, while hLf has been shown to be absent from lung alveoli [21,31]. These species-specific differences in the chemical properties and physiological distribution may indicate functional distinctions; however, both hLf and bLf exhibit an array of characteristics during infection.

During viral infection, Lf directly impacts the host immune response through the rebalancing of iron homeostasis, the modulation of inflammation, and the promotion of antiviral gene expression [32]. For example, the oral administration of bLf has been shown to reduce levels of IL-6 systemically, leading to decreased hepcidin and increased ferroportin [21]. As described above, this leads to the increased release of intracellular iron stores and dietary uptake of iron to counter inflammation-induced anemia. More passively, Lf directly binds free iron, thereby limiting the iron-dependent inflammatory processes in tissues [33].

There are many mechanisms by which Lf reduces inflammation, including both active and passive processes. For example, the in vitro culture of human macrophages with bLf has also been shown to play a role in shifting LPS-stimulated, pro-inflammatory M1 macrophages to the M2 anti-inflammatory phenotype [32]. Whether this shift is through the direct modification of gene expression or through more passive means of metabolic immunomodulation remains to be determined; however, multiple studies have shown that both hLf and bLf are transported into the nucleus and can alter gene expression [34,35,36] (Figure 2B). Indeed, Lf is proposed to function as a transcription regulator and to directly inhibit the expression of pro-inflammatory cytokines [37]. For example, Ashida et al. demonstrated that differentiated intestinal epithelium-like Caco-2 cells bound and internalized endogenously expressed Lf at the apical membrane, eventually localizing Lf to the nucleus [36]. In another in vitro study using peripheral blood-derived human mononuclear cells (PBMCs), bLf was rapidly brought into the nucleus during monocyte cellular differentiation, correlating the differentiation with bLf reaching the nucleus [34]. Using primary bronchial cells derived from cystic fibrosis patients, Valenti et al. demonstrated that the addition of bLf during infection reduced inflammation through decreased pro-inflammatory cytokine expression and increased anti-inflammatory IL-10 expression (Figure 2B). Interestingly, the same study also asserted that bLf only altered the expression of cytokines in infected cells and not in uninfected cells [38]. Finally, using human umbilical vascular endothelial cells (HUVECs), Kim et al. demonstrated in vitro that hLf interfered with the TNF-α-induced expression of intracellular adhesion molecule-1 (ICAM-1) by downregulating *ICAM-1* expression through a DNA-binding-dependent manner [35] (Figure 2B). Clearly, there is mounting evidence suggesting that Lf, of both bovine and human origin, can actively modulate gene transcription, impacting both iron homeostasis and the inflammatory process.

### 3.2. Antiviral Activity of Lactoferrin

The antiviral activity of lactoferrin has been demonstrated both in vitro and in vivo (summarized in Table 1). This antiviral property was first demonstrated in mice infected with the polycythemia-inducing strain of the friends virus complex (FVC-P). For this study, mice were intravenously dosed with hLf, which prolonged their survival rates and decreased the viral load of FVC-P [39]. Lf is able to inhibit viruses, in part, through iron chelation [2]; however, subsequent studies indicated that the antiviral mechanism of lactoferrin is much more complex. Due to its cationic features and its ability to bind iron at a low pH, Lf is able to interact with many surface molecules and ions, including heparan sulfate proteoglycans, cell receptors, and enveloped viral particles, leading to the disruption of viral maturation and inhibition of immunomodulatory activation [2,40,41].

Despite this diverse functional range, certain factors have been shown to impact Lf’s efficacy in limiting viral infection. For example, the evidence from multiple in vitro studies suggests that the antiviral effects of Lf are particularly effective early and are mediated primarily through the prevention of viral entry into host cells [2,52,53]. In addition, the antiviral efficacy may be impacted by subtle differences between species, with bLf reported to exhibit higher antiviral activity than hLf [2,42]. For example, Marchetti et al. showed that both hLf and bLf have the ability to interfere with herpes simplex virus-1 (HSV-1) infection by binding heparin sulfate proteoglycans and LDL receptors (Figure 2A); however, bLf was shown to be more potent in this case [43]. The specific biochemical or structural differences that contribute to Lf’s functional divergence between primates and bovines remains to be established; however, the species of origin may impact Lf’s antiviral efficacy.

The growing consensus also indicates that Lf’s antiviral activity levels are not identical between infecting virus types [44,45,46,71]. While the above studies indicate that Lf’s antiviral activity is mediated through the competitive inhibition of viral binding and entry into the host cell, other studies have demonstrated Lf’s inhibition of viral-infection-mediated post viral attachment. For example, in an in vitro study examining both human and bovine Lf inhibition of poliovirus replication, Manchetti et al. demonstrated that both hLf and bLf prevented the viral replication of poliovirus, regardless of whether iron, magnesium, or zinc was bound; however, the post-attachment inhibition of viral infection was dependent on zinc-bound Lf, with the levels of zinc and degree of inhibition being directly correlated [45]. Another in vitro study examined the role of bLf in limiting rotavirus infection and found that the removal of sialic acid enhanced the anti-rotavirus activity of bLf, concluding that the mechanism of viral suppression in rotavirus infection is not dependent on the competitive inhibition of viral attachment, as shown for HSV infection [46].

In some circumstances, Lf may exert a variety of antiviral mechanisms to prevent infection. In both in vitro and in vivo studies using pull down assays, the role of Lf in hepatitis-C virus (HCV) infection has shown that Lf directly binds the HCV envelope protein E2 (Figure 2A), rather than competitively inhibiting HCV attachment to the host cell [44,47]. In addition, further in vitro studies on HCV and Lf showed that HCV leads to elevated intracellular iron stores. This elevation of iron levels increased the susceptibility to infection by the virus, suggesting that the iron sequestering role of Lf also helps limit the viral pathogenesis [14]. Lf has also been shown to prevent hepatitis-B virus (HBV) infection; however, in contrast to HCV infection, the proposed mechanism is through the competitive binding of the glycosaminoglycan receptor [48] (Figure 2A), suggesting that Lf antiviral mechanisms may vary even within closely related viruses.

These promising in vitro and in vivo studies, as well as the GRAS status of Lf, have led to an increased interest in the clinical use of Lf in the treatment of viral infections (systematically reviewed by Sinopoli et al. [6]). Using a standardized approach [72,73], Sinopoli et al. identified 27 records investigating the use of orally administered Lf to prevent or manage viral infections (Table 1). The viral infections studied included HCV, HIV, rotavirus, norovirus, and SARS-CoV-2, and ranged from non-randomized to randomized, dose–response, and controlled trials. Minimal side effects were identified across all studies, and the treatment impacts varied from no observed differences to significant impacts on the viral load, immune response, and reported symptoms. For example, the management of chronic HCV patients with oral bLf resulted in significant decreases in alanine transaminase and plasma 8-isoprostane and significantly increased IL-18, CD4, CD8, CD137, and CD56 levels [49,50,51,55]. For norovirus, rotavirus, and SARS-CoV-2, decreased duration and severity of symptoms were observed [56,57,65,67]. Finally, decreases in the viral load and duration of viral detection were observed for both HIV and SARS-CoV-2 [58,65]. While these studies provide promising results, establishing oral Lf as a therapeutic option for the management of viral infections will require larger and more robust clinical trials.

### 3.3. Lf as a Therapeutic Adjuvant in COVID-19

Since the emergence of the coronavirus disease 2019 (COVID-19), caused by severe acute respiratory syndrome coronavirus 2 (SARS-CoV-2), it has effectively spread across the globe, with more than 529 million people confirmed to have been infected and more than 6.3 million deaths by the spring of 2022 (the Johns Hopkins Coronavirus Resource Center: https://coronavirus.jhu.edu, accessed on 1 June 2022). Even with the successful development of vaccines, much of the population remains susceptible to the virus due to such factors as a lack of access to vaccines, hesitancy towards vaccination, and the emergence of viral variants. In addition, the long-term impacts of COVID-19 have led to the emergence of chronic pathologies referred to as “ongoing symptomatic COVID-19” for symptoms lasting between four and twelve weeks post-acute infection and “post-COVID-19 syndrome” for symptoms lasting longer than twelve weeks [74]. With the continued spread of COVID-19, and despite the development of SARS-CoV-2 vaccines, there remains the need to develop novel therapeutics for both acute and chronic COVID-19.

Disrupted iron homeostasis has been associated with worse outcomes in viral infections, as previously mentioned, and the same is seemingly true for patients with COVID-19. The accumulating evidence suggests that iron chelation therapy is a promising adjuvant therapy for COVID-19 [75]. While the mechanism of action is still under debate, COVID-19 is known to be associated with increased levels of the proinflammatory factors IL-1β, IFNγ, IP-10, and MCP-1, the expression of which is sensitive to iron homeostasis [76]. Additionally, macrophages are presumed to be infected by COVID-19, and increased iron storage in macrophages may favor viral replication [77]. More recently, Cutone et al. demonstrated in both enterocytes and macrophages that the SARS-CoV-2 spike protein can induce the dysregulation of major iron-handling proteins, including the downregulation of ferroportin, DMT-1, and hephaestin and the upregulation of TfR1 [66].

Patients with severe COVID-19 infections tend to have elevated ferritin and IL-6 levels, with the serum ferritin levels demonstrated to be twice as high in non-survivors [78]. High serum ferritin levels are linked to cardiovascular events in addition to inflammatory pathologies, and cardiac damage is a common outcome of severe COVID-19 [79]. Ferroptosis, a form of programmed cell death mediated by iron, is initiated by alkoxyl radicals produced by ferrous iron [80]. Ferroptosis has been shown to be linked to inflammation, as it involves multiorgan inflammatory pathways in the liver, kidney, heart, and lungs [81,82,83]. Observed in both adult COVID-19 and the pediatric multisystem inflammatory syndrome in children (MIS-C), the inflammatory damage of the host tissue can occur in the most severe cases of SARS-CoV-2 infection [84]. Interestingly, ferroptosis is also associated with ageusia and anosmia (loss of taste and smell, respectively). Hypogeusia, a reduction in tasting ability, and reduced olfactory function were associated with iron deficiency in patients [85,86]. Just as with iron deficiency and ferroptosis, COVID-19 has also been shown to be associated with anosmia and ageusia [87].

Another known risk factor and major concern for COVID-19 patients is the disruption of the coagulation cascade. Iron plays a complex role in coagulation, extending the clotting of plasma by interacting with proteins of the coagulation cascade, while increasing the risk of thrombosis by precipitating plasma proteins [88]. COVID-19 patients have already been shown to be at an increased risk for thromboembolism. In fact, the preliminary reports on COVID-19 patients show that the patients can often present with thrombocytopenia, elevated D-dimer levels, prolonged prothrombin time, and disseminated intravascular coagulation, along with coagulation cascade abnormalities [89]. Other studies have touched on the inflammatory and procoagulant condition of COVID-19 patients, which is especially prominent in patients with less-favorable outcomes [90]. One study with five patients under the age of 50, all with confirmed SARS-CoV-2 infection, reported that each patient experienced new-onset large-vessel strokes [90]. Similarly, another study examined patients in a COVID ICU, where clinically significant coagulopathy and multiple infarcts were seen [79]. From an excessive proinflammatory state to multiorgan oxidative damage to anosmia and ageusia, the pathologies associated with COVID-19 indicate that beyond a characteristic hyperferritinemia, this disease exhibits many features of the systemic dysregulation of iron homeostasis.

### 3.4. Antiviral Activity of Lactoferrin against SARS-CoV-2

Because of its ability to modulate viral infection, iron homeostasis, and inflammation, Lf warrants consideration as an adjuvant therapy in the treatment of COVID-19. In particular, the use of dairy-derived lactoferrin has been shown to be efficacious in vitro and in vivo, can be economically produced in large quantities, and is well tolerated in clinical trials. Over the course of the pandemic, suggestive data supporting this therapeutic approach have emerged. Mirabelli et al. used quantitative high-content morphological profiling coupled with AI-based machine learning to screen 1425 FDA-approved compounds and clinical candidates, and found bovine lactoferrin to be an effective inhibitor of SARS-CoV-2 infection. The mechanistic basis for this inhibition was not clear; however, they showed that Lf blocks SARS-CoV-2 viral entry and rescues infection up to 24 h post-infection in a dose-dependent reduction in viral replication. In addition, they observed elevated mRNA levels of IFN-β and associated IFN genes (*ISG15*, *MX1*, *Viperin*, and *IFITM3*) in Huh7 cells [59]. Subsequently, Lf was shown in vitro to inhibit SARS-CoV-2 infection and replication in Caco-2 intestinal epithelial cells, with an associated increased expression of antiviral genes (*IFNA1*, *IFNB1*, *TLR3*, *TLR7*, *IFIH1*, *IRF3*, *IRF7*, and *MAVS*) and decreased expression of the anti-inflammatory cytokine *TGFB1* [60]. More recently, Wotring et al. screened H1437 human lung cells cultured in vitro and pre-treated with a series of dairy products against several SARS-CoV-2 emergent variants of concern (B.1.1.7, B.1.351, and P.1), finding that the antiviral efficacy of the dairy products was concentration-dependent on bLf [62]. These in silico and in vitro results further support the potential of bLf as a nutraceutical protein for treating SARS-CoV-2 infection.

How Lf can inhibit SARS-CoV-2 viral entry is currently under investigation, but the likely mechanisms include directly binding host cell factors (competitive inhibition) or directly binding viral particles (Figure 2), as previously described for other viruses. While bLf has been shown to directly bind viral structural proteins such as S, M, and E [45], a number of studies have suggested that Lf may directly interfere with the viral entry into host cells through the competitive inhibition of viral binding to glycan attachment factors. The two key classes of glycans include the sialic acid glycans and glycosaminoglycans, such as heparan sulfate. While the angiotensin-converting enzyme (ACE2) receptor is a well-established binding target for SARS-CoV-2 [91], glycans are proposed to initiate contact with viral particles, shuttling the virus to the target binding receptor on the host membrane and facilitating viral engulfment [92]. Both sialic acid and heparan sulfate glycan receptors have been shown to be involved in SARS-CoV-2 infections [93,94,95,96], and Lf may block SARS-CoV-2 viral entry by binding these accessory targets to viral entry. The computational modeling of molecular interactions demonstrated putative regions on the surface of Lf that could potentially competitively inhibit the viral attachment to sialic acid and the complementary surface structure on the SARS-CoV-2 spike protein, potentially allowing Lf to compete with ACE2 receptors for binding directly to viral particles [63].

More recently, Piacentini et al. used bilayer interferometry and latex nanoparticle-enhanced turbidity to measure both the kinetic and thermodynamic parameters of hLf binding to the SARS-CoV-2 receptor-binding domain (RBD) and the ACE2 receptor. Their results indicated that hLf binds the ACE2 receptor but not RBD at physiological concentrations of hLf. Interestingly, they found that Lf bound the ACE2 receptor ectodomain, indicating that hLf may competitively inhibit RBD binding by either directly interfering with the ACE2-RBD interaction or by binding an ACE2 receptor region far from the RBD site, triggering conformational changes of the ACE2 receptor and inhibiting RBD binding [64]. In contrast, Cutone et al. recently showed that both hLf and bLf do interact with the SARS-CoV-2 trimeric form of a full-length spike protein, even with the introduction of point mutations observed in recent viral variants [66]. Clearly, further studies are warranted; however, these in vitro results are promising, especially given the continued efficacy of Lf when challenged with newly emergent viral variants.

### 3.5. Clinical Evidence of Lactoferrin Efficacy in COVID-19 Patients

Clinically, the evidence for the use of Lf as a therapeutic adjuvant for SARS-CoV-2 infection is limited but rapidly increasing. Campione et al. [65] performed a small (*n* = 32) human trial on bLf embedded in liposomes for oral and nasal administration in asymptomatic to mild symptom COVID-19 cases. This treatment resulted in early viral clearance (15–30 days) along with decreases in IL-6, D-Dimer, and ferritin serum levels. The in silico modeling suggested the direct binding of Lf to S protein in the study. While these results are indeed promising, it is important to note that there was no placebo control group in the study, nor was it a double-blind clinical trial [65]. Another pilot clinical trial (*n* = 54) using oral bLf for mild to moderately symptomatic COVID-19 patients was randomized and assayed for dose-dependent responses. While no statistically significant differences between the control and treatment groups were observed, some interesting trends were noted. For example, the reported symptoms of fever, dry cough, diarrhea, headache, loss of sense of taste or smell, and fatigue were improved with Lf treatment. Increased serum hemoglobin, lymphocyte count, and platelet count levels were detected at the highest dose of Lf (200 mg, twice daily), while the levels of C-reactive protein declined [69]. Finally, in a retrospective study on asymptomatic, paucisymptomatic, and moderately symptomatic COVID-19 patients in home-based isolation, Rosa et al. compared Lf-treated (*n* = 82) to Lf-untreated (*n* = 39) patients using bLf unloaded in liposomes. The median number of days to negative molecular test for the SARS-CoV-2 virus was lower in bLf-treated patients compared to non-treated patients (15 vs. 24 days, respectively); however, the times to symptom resolution did not differ significantly, with the exception of a subcohort of individuals of advanced age [70]. While these preliminary clinical trials are promising, much larger randomized studies are needed to establish the clinical value of Lf supplementation.

Finally, each of these clinical trials utilized oral Lf supplementation (with the exception of Campione et al. [51], who also used a nasal spray of liposomal Lf). Orally administrated Lf is subject to digestion by gastrointestinal proteases and can be absorbed as amino acids via facultative diffusion or active transport; however, Lf given orally interacts with receptors on gut epithelial cells and is likely to have systemic effects through signaling pathways. While Wotring et al. [88] did assay the effect of lactoferricin f17-41, which is produced during digestion, against SARS-CoV-2 infection in vitro, they found this bLf derivative to be less effective than whole bLf. Clearly, much remains to be understood about Lf as a functional dietary supplement in general and as a therapeutic adjuvant against SARS-CoV-2 specifically.

## 4. Conclusions

In conclusion, iron homeostasis is important for health, as iron levels play key roles during viral infections. Bovine lactoferrin’s ability to modulate iron levels both systemically and locally, to mediate inflammatory processes, and to directly inhibit viral entry into host cells make it a potential therapeutic option for patients infected with SARS-CoV-2. As described above, bLf is structurally and functionally similar to hLf, can be effective when administered orally, is easily tolerated, and can be economically produced. These characteristics of dairy-derived bLf make it a viable candidate for further studies, specifically as a dietary supplement that is accessible to under-resourced populations impacted by the SARS-CoV-2 pandemic. While in many parts of the country the pandemic is still in the acute phase of its spread, Lf is also likely to have long-term value as we enter the post-vaccination phase of the pandemic. Approximately 30% of COVID-19 patients are reporting persistent symptoms lasting longer than nine months, despite having resolved the acute phase of their illness [97]. The characteristics of post-acute sequelae of COVID-19 (PASC), otherwise referred to as long COVID, include fatigue, muscle weakness, insomnia, palpitations, chronic rhinitis, dysgeusia, chills, sore throat, and headache [98]. Many of these symptoms are associated with the chronic, systemic inflammation and dysregulation of iron homeostasis, targets for which bLf is an established therapeutic option. While much remains to be understood about the therapeutic benefits of Lf as an adjuvant for SARS-CoV-2, the potential benefits of further investigations are likely to yield results both in the short-term pandemic surge and the longer-term outcomes of persistent symptoms in PASC patients. As the SARS-CoV-2 pandemic shifts towards endemic and potentially seasonal patterns of emergence, more clinical trials that examine the efficacy of oral bLf as a prophylactic for immune priming, a therapeutic adjuvant in acute infection, and a long-term nutraceutical for symptoms of PASC will provide the evidence needed to add bLf to our therapeutic toolbox in the fight against emerging viral pathogens such as SARS-CoV-2.

## Figures and Tables

**Figure 1 nutrients-14-03090-f001:**
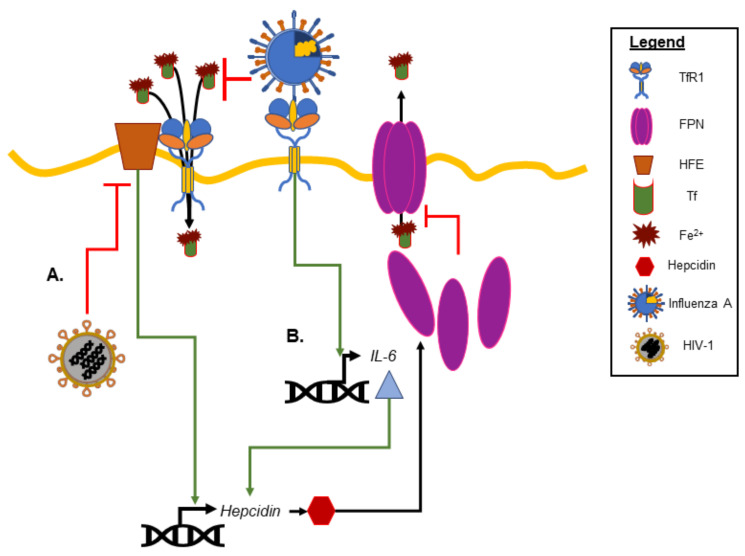
**Examples of iron homeostasis and viral infection.** (**A**) Human immunodeficiency virus (HIV-1) blocks the homeostatic iron regulator (HFE) to increase iron accumulation in macrophages. (**B**) Influenza A binding of the transferrin receptor 1 (TfR1) promotes the expression of IL-6 and hepcidin degradation of ferropotin (FPN), also leading to intracellular iron retention and the promotion of viral replication.

**Figure 2 nutrients-14-03090-f002:**
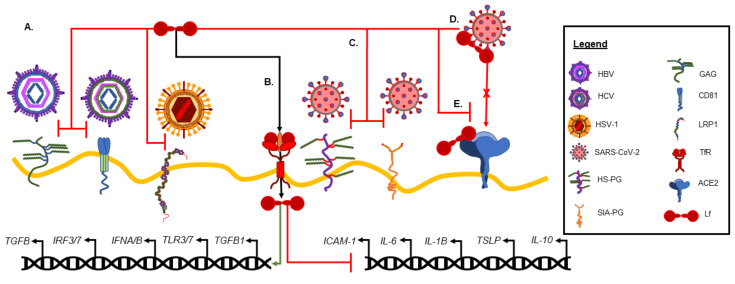
Lactoferrin and viral infection. (**A**) Lactoferrin (Lf) can directly prevent viral entry through preventing viral engagement with host cell target receptors, such as hepatitis B virus (HBV) binding to glycosaminoglycan (GAG), hepatitis c virus (HCV) binding to cluster of differentiation 81 (CD81), and herpes simplex virus type 1 (HSV-1) binding to low-density lipoprotein receptor-related protein 1 (LRP1). (**B**) Lf uptake through the transferrin receptor (TfR) by host cells promotes the expression of antiviral genes and the inhibition of proinflammatory genes. (**C**) The proposed Lf mechanism in SARS-CoV-2 infection includes blocking viral binding to accessory target molecules, such as heparan sulfate proteoglycan (HS-PG) and sialic acid glycoprotein (SIA-PG), (**D**) direct binding to viral particles, and (**E**) the competitive inhibition of viral binding to the angiotensin-converting enzyme (ACE2) receptor. Additional abbreviations include: TGFB = transforming growth factor beta; IRF = interferon regulatory transcription factor; IFN = interferon; TLR = toll-like receptor; ICAM = intracellular adhesion molecule; IL = interleukin; TSLP = thymic stromal lymphopoietin.

**Table 1 nutrients-14-03090-t001:** Summary of antiviral studies with lactoferrin. In silico, in vitro, and clinical trials utilizing human, bovine, or other sources of lactoferrin are outlined below with the source of lactoferrin, route of administration (if in vivo), and summary of the relevant results from the main text.

Author (Year) [Citation]	Model	Lactoferrin Source Route (Dose)	Brief Results
**Sinopoli et al. (2022) [6]**	Systemic Review	NA	Systemic review of clinical trials using orally administered Lf for the treatment of viral infections.
**Marchetti et al. (1996) [42]**	Primate In vitro	hLf bLf	Lf inhibits HSV1 absorption with bLf showing better efficacy than hLf.
**Lu et al. (1987) [39]**	Murine In vivo	hLf i.p.	hLf shown to have protective effects against the polycythemia-inducing strain of the friend virus complex in mice.
**Marchetti et al. (1998) [43]**	Primate In vitro	bLf	The antiviral activity of Lf appears to correlate with the degree of its metal binding and saturation.
**Yi et al. (1997) [44]**	In vitro	bLf hLf	Demonstrates interaction of Lf and HCV envelope proteins.
**Marchetti et al. (1999) [45]**	Primate In vitro	bLf	Suggests bLf plays a role in altering viral infection, particularly in the gut, through the inhibition of certain steps of viral infection.
**Superti et al. (2001) [46]**	Primate In vitro	bLf	bLf inhibits rotavirus through a different mechanism than the previously reported for HPV.
**El-Fakharany (2013) [47]**	Human In vitro	hLf bLf camel Lf sheep Lf	Human, camel, bovine, and sheep Lf prevent HCV entry into cells by binding the virus; camel Lf was most effective.
**Hara et al. (2002) [48]**	Human In vitro	bLf hLf	Lf inhibits HBV infection in vitro.
**Ishii et al. (2003) [49]**	Human Clinical	bLf oral (0.6 g/day)	Increased IL-18 with oral bLf supplement in chronic HCV patients.
**Okada et al. (2002) [40]**	Human Clinical	bLf oral (1.8–7.2 g/day)	bLf use in chronic hepatitis C patients is well tolerated.
**El-Ansary et al. (2016) [50]**	Human Clinical	bLf oral (0.5 g/day)	Increased CD4, CD8, CD137, and CD56 levels with bLf supplementation in chronic HCV patients
**Ueno et al. (2006) [41]**	Humans Clinical	bLf oral (1.8 g/day)	Oral Lf has a negligible impact on viral load when taken orally by patients with chronic HCV.
**Tanaka et al. (1999) [51]**	Humans Clinical	bLf oral (1.8–6 g/day)	Lf could be used as an anti-HCV adjuvant therapy with the potential to help treat chronic hepatitis.
**Hirashima et al. (2004) [52]**	Human Clinical	bLf oral (9.0 g/day)	Lf did not increase the response rate or prevent relapse after discontinuing interferon in chronic HCV patients.
**Ishibashi et al. (2005) [53]**	Human Clinical	bLf oral (0.6 g/day)	This study failed to demonstrate that Lf in combination with antiviral therapy provided additional benefit to chronic HCV patients.
**Kaito et al. (2007) [54]**	Human Clinical	bLF oral (3.6 g/day)	Lf was shown to increase the effectiveness of interferon and ribavirin therapy in chronic HCV patients.
**Konishi et al. (2006) [55]**	Human Clinical	bLf oral (3.6 g/day)	Decreased ALT levels and plasma 8-isoprostane in chronic HCV patients.
**Ochoa et al. (2013) [56]**	Human Clinical	bLf oral (0.5 g/day)	Decreased duration and symptoms in norovirus patients.
**Egashira et al. (2007) [57]**	Human Clinical	bLf oral (100 mg/day)	Decreased frequency and duration of symptoms in rotavirus patients.
**Zuccotti et al. (2007) [58]**	Human Clinical	bLf oral (3 g/day)	Observed decline in viral load during bLf administration in HIV patients.
**Mirabelli et al. (2020) [59]**	Human Primate In vitro	bLf hLf	Lf effective, in vitro, at inhibiting COVID through multiple mechanisms.
**Salaris et al. (2021) [60]**	Human Primate In vitro	bLf	Lf-moderated immunity during SARS-CoV-2 infection.
**Oda et al. (2021) [61]**	Human In vitro	bLF	bLf demonstrates antiviral activity against the human norovirus
**Wotring et al. (2022) [62]**	Human In vitro	bLf	Dairy product efficacy in inhibiting SARS-CoV-2 infection was dependent on Lf concentration; bLf retained efficacy against SARS-CoV-2 viral variants of concern.
**Miotto et al. (2021) [63]**	In silico	hLF	Computational modeling indicated that Lf blocks SARS-CoV-2 infection through competitive binding with the spike protein.
**Piacentini et al. (2022) [64]**	In silico	hLf	Lf binds to ACE2 receptor and not SARS-CoV-2 spike protein RBD.
**Campione et al. (2021) [65]**	Human Primate In vitro	bLf	Lf effective antiviral against SARS-CoV-2 infection in vitro.
**Cutone et al. (2022) [66]**	Human In vitro	bLf	Preincubation with bLf inhibited SARS-CoV-2 binding and pseudovirus entry into epithelial and macrophage-like cells, reduced inflammatory response, and increased gene expression associated with iron homeostasis.
**Serrano et al. (2020) [67]**	Human Clinical	bLf oral (20–30 mg/day)	Improvement in reported symptoms in mild to moderate COVID-19 patients.
**Campione et al. (2020) [68]**	Human Clinical	bLf oral (1 g/day)	Decreased time to negative molecular test and duration of symptoms in COVID-19 patients
**Algahtani et al. (2021) [69]**	Human Clinical	bLf oral (200–400 mg/day)	No statistical difference between treatment and non-treatment groups, but trends in symptom improvement and blood biomarker profile observed.
**Rosa et al. (2021) [70]**	Human Clinical	bLf oral (200–1000 mg/day)	Reduced time to negative molecular SARS-CoV-2 test, reported reduction in symptoms of COVID-19 patients of advanced age.

Abbreviations: NA = Not applicable, bLf = bovine lactoferrin, hLf = human lactoferrin, i.p. = intraperitoneal.

## Data Availability

Not applicable.

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
