# Peer review of "The Influence of Viral Infections on Iron Homeostasis and the Potential for Lactoferrin as a Therapeutic in the Age of the SARS-CoV-2 Pandemic"

_nutrients, 2022, doi:10.3390/nu14153090_

Round 1

Reviewer 1 Report

The Manuscript entitled: "Iron Homeostasis, Anemia of Inflammation, Lactoferrin, and SARS-CoV-2 Infection" by Ward et al. describes the iron and inflammatory homeostasis disorders induced by SARS-CoV-2 and the potential role of lactoferrin in counteracting viral infection as well as rebalance the disorders induced by inflammation.

The Manuscript is very confused. In particular, the paragraphs on "iron homeostasis" and "iron homeostasis and lung injury" lack an explanation of iron homeostasis's molecular mechanisms in health and diseases, with a particular emphasis on the major proteins involved in iron homeostasis.

Please, add in the paragraph "iron homeostasis and lung injury" the different modulation of the major proteins of iron homeostasis in health and diseases with a particular concern on the ferroortin localization and function. 

In the paragraph on "the antiviral activity of lactoferrin" Table 1 refers to the studies carried out on the antibacterial activity of lactoferrin. In addition, Table 1 lacks a caption.

The paragraph "Lf as therapeutic adjuvant in COVID-19" should be divided in "Anviviral activity of Lf against SARS-CoV-2" in which the authors report the in vitro studies, and in the second paragraph in which the Authors report the in vivo studies. About the last point, in lines 447-453, the authors report the in vivo studies by Campione E. et al. “Lactoferrin as Antiviral Treatment in COVID-19 Management: Preliminary Evidence.” International journal of environmental research and public health vol. 18,20 10985. 19 Oct. 2021, doi:10.3390/ijerph182010985 but the reference reported refers to the in vitro study of the same group (ref. 92). 

Other references have been reported, such as:

-Piacentini, R et al. “Lactoferrin Inhibition of the Complex Formation between ACE2 Receptor and SARS CoV-2 Recognition Binding Domain.” International journal of molecular sciences vol. 23,10 5436. 13 May. 2022, doi:10.3390/ijms23105436

Wotring, Jesse W et al. “Evaluating the in vitro efficacy of bovine lactoferrin products against SARS-CoV-2 variants of concern.” Journal of dairy science vol. 105,4 (2022): 2791-2802. doi:10.3168/jds.2021-21247

Rosa, L et al. “Ambulatory COVID-19 Patients Treated with Lactoferrin as a Supplementary Antiviral Agent: A Preliminary Study.” Journal of clinical medicine vol. 10,18 4276. 21 Sep. 2021, doi:10.3390/jcm10184276

Algahtani, FD et al. “The Prospect of Lactoferrin Use as Adjunctive Agent in Management of SARS-CoV-2 Patients: A Randomized Pilot Study.” Medicina (Kaunas, Lithuania) vol. 57,8 842. 19 Aug. 2021, doi:10.3390/medicina57080842

Cutone A, et al. Lactoferrin binding to Sars-CoV-2 Spike glycoprotein protects host from infection, inflammation and iron dysregulation., 17 May 2022, PREPRINT https://doi.org/10.21203/rs.3.rs-1605740/v1

- Table 1 should be reviewed by including only the studies on the antiviral activity of lactoferrin

- Please, add the concentration of lactoferrin used in the in vitro and in vivo studies.

- Note any typing errors

- Lines 172-173: Please, add the reference

Author Response

Reviewer 1:

              We thank the Reviewer for their insightful review, comments, and suggestions for revision of our manuscript. Based on this review and others, we have substantially revised the manuscript and hope Reviewer 1 finds it much improved. Specifically, we hope that by reorienting the focus and narrowing the scope, this revised version is more readable and informative.

              All three reviewers felt that the paragraphs on “iron homeostasis” and “iron homeostasis and lung injury” needed substantial revision. Specific to Reviewer 1 comments, we have revised the iron homeostasis discussion to be more focused on the major proteins involved in iron homeostasis which are involved or impacted during viral infection, as this is the focus of this manuscript. Based on feedback from all three reviewers, we have removed the paragraph “iron homeostasis and lung injury”, as the reviewers felt the paragraph was out of scope for the manuscript and only added to their confusion.

              For the paragraph on “the antiviral activity of lactoferrin”, we appreciate Reviewer 1 recommending removal of antibacterial references and agree these studies are outside of the scope of the manuscript. We have substantially revised both Table 1 and the associated paragraphs. In addition, we have divided the “lactoferrin as a therapeutic adjuvant in COVID-19” into sections discussing the disrupted iron and inflammatory homeostasis in COVID-19, in vitro and in silico studies of lactoferrin with SARS-CoV-2, and clinical studies of lactoferrin and COVID-19. We appreciate Reviewer 1 suggesting this change and hope it makes this section more readable and informative. Finally, we also corrected in vitro/ in vivo reference mistake for the two studies by Campione et al. and thank Reviewer 1 for catching this error.

              We have added the additional references recommended by Reviewer 1 and very much appreciate Reviewer 1 for bringing these studies to our attention. We feel the addition of these references greatly improves the manuscript and our hypothesis that lactoferrin has value as a therapeutic in COVID-19.

              Finally, minor modifications requested by Reviewer 1 have been addressed. Specifically, concentration has been added to Table 1, typing errors have been addressed, and reference has been added to GRAS designation.

              Again, we thank Reviewer 1 for their comprehensive review and believe this revised manuscript is improved by incorporation of their suggestions and comments.

Reviewer 2 Report

The subject matter is of high interest but while much information has been collected, it is not well synthesized, and our understanding has not been much improved. The review is very difficult to read, covers a much too wide area and speculates too much when the exact mechanisms are not clearly known or understood. It would be better to focus on how viral infections alter iron homeostasis and how bovine lactoferrin may be useful as a therapeutic adjuvant in viral respiratory infections.

A major error is assuming that high serum ferritin levels are due to high iron stores when, in most of the examples discussed, they are due to inflammation as serum ferritin is an acute phase protein.

The title is misleading. The review deals mainly with how viral infections influence iron homeostasis and how lactoferrin may help therapeutically to treat viral infections including SARS CoV-2. The anemia of inflammation does not need to be in the title and SARS CoV-2 presumably is added to increase interest. A better title could be “The influence viral infections on iron homeostasis and the usefulness of bovine lactoferrin and as a therapeutic agent.

Lines 44-101 is less relevant to the main subject, includes speculation and could be omitted, or included in a much-condensed form in an introduction.

Specific points.

1.If the abstract is kept then the text needs to be focused to meet the abstract.

2.Lines 46-52. I do not understand the relevance of the distribution presented, please explain.

3.Lines 51-2. This is unclear. Do you mean that iron entering into the body via the lung could impact on iron absorption from the intestine and body iron homeostasis? This seems extremely unlikely as the amount is so low that it would not influence iron status. It is possible that the airborne iron particles irritate the lung surface and cause inflammation. Such an effect could occur with many other particles and would not be specific to iron. Maybe the lines 44-60 could be summarized and integrated into the following section.

4.P.61 The following section (iron homeostasis and lung injury) is very heterogenous and sub titles would help the reader eg; inhaled iron, ARDs following ischemic or perfusion injury, smoking, and cystic fibrosis.

5.P74 What is meant by “increased iron exposure can exacerbate ARDS ”? Do you mean do you mean more iron particles entering the lung with air, increased iron intake through food or supplements (more meat), hemochromatosis? It would help is you could state that there is no evidence that increased Fe intake through food or supplements exacerbate ARDS and that the lung damage is due to physical injury. A better title for the section would be that lung injury alters iron homeostasis and then discuss whether it is lung iron homeostasis or whole-body iron homeostasis.

6.Lines 75-76. The high serum ferritin is most likely due to inflammation not high iron stores. It has been a common mistake to propose high iron stores as a risk factor for cardiovascular disease and cancer. Serum ferritin is an acute phasr protein and increases with inflammation.

7.Line 90. Please describe the form of extracellular iron and state what is known of its origin. Additionally, mention that the increases in SF could be due to inflammation.

8.Lines 97-98. I do not thing you have provided enough evidence that iron is elevated in many lung disorders (not ARDS or smoking?). You have described changes in iron homeostasis. Free iron could be measured by the technique of non-transferrin bound iron.

9.Line 112 What evidence is there that viral infections increase circulating or intracellular free iron?

10. Line 115. Does excessive inflammatory response increase free iron?

11.Lines 116-144 contains much information which could be better synthesized.

12.Line 148 Is it only the binding of influenza virus A to TfR that increases IL6. This seems very unlikely.

13.Fig 1 is very ornate but only covers 2 of many influences of viral infection on Fe homeostasis so is not so useful.

14.The rest of the text contains much information on the influence of viral infections on iron homeostasis and the use of lactoferrin as a therapeutic agent and could form the basis of a useful review if synthesized and presented in a more readable way.

Author Response

Reviewer 2:

We thank the Reviewer for their insightful review, comments, and suggestions for revision of our manuscript. We appreciate Reviewer 2 comment that the manuscript contains subject matter of high interest and contains informative value. Based on this review and others, we have substantially revised the manuscript and hope Reviewer 2 finds it much improved. Specifically, we hope that by reorienting the focus and narrowing the scope, this revised version is more readable and informative. We appreciate Reviewer 2 knowledge of iron homeostasis and inflammation. In particular, we hope we have adequately addressed errors in the text inaccurately indicating that high serum ferritin levels are due to high iron stores, rather than due to inflammation. We have revised the manuscript title based on Reviewer 2 recommendations and hope it more accurately reflects the revised and more focused scope of the revised manuscript. In addition, we have retained the abstract, but refocused the main body of the manuscript to better reflect the abstract, as recommended by Reviewer 2.

              All three reviewers felt that the paragraphs on “iron homeostasis” and “iron homeostasis and lung injury” needed substantial revision. Based on feedback from all three reviewers, we have removed the paragraph “iron homeostasis and lung injury”, as the reviewers felt the paragraph was out of scope for the manuscript and only added to their confusion. In addition, Reviewer 2 accurately indicated specific over interpretation by us, as well as speculation and we agree the section is better omitted from the manuscript.

For the paragraph on “iron homeostasis”, we have condensed and reoriented the focus on elements of iron homeostasis relevant to viral infection. Specific to Reviewer 2 comments, we have included additional clarification about viral infections and the impact on iron homeostasis and inflammation. We have also included a reference to an excellent review that comprehensively outlines the many processes that contribute to viral disruption of iron homeostasis, as we feel a comprehensive review of this topic is outside of the scope of this review; however, Reviewer 2 brings up some very interesting questions about circulating and intracellular free iron, viral infection, and excessive inflammation. Finally, we agree with Reviewer 2 that during viral infection there are many mechanisms contributing to elevated IL-6 increase; however, specific to viral mechanisms of direct enhancement of the inflammatory response, the engagement of influenza A virus with the TfR has been shown to directly trigger increased gene expression of IL-6, thus increasing circulating IL-6 and the inflammatory state of the host.

While we acknowledge and can appreciate Reviewer 2 comment on Figure 1, we did wish to visualize at least a couple of examples of viral infection impacting iron homeostasis and inflammation. Neither of the other reviewers recommended removing the figure, so we respectfully request to keep it in this revised manuscript.

Finally, we appreciate Reviewer 2 kind comments on the value of the information presented in this review manuscript and hope that with this revision Reviewer 2 will find this version much more readable, informative, and valuable.

Reviewer 3 Report

Thank you for the opportunity to revise this manuscript.

Recent studies indicate that a hyperinflammatory syndrome induced by SARS-CoV-2 contributes to disease severity and mortality in COVID-19. In particular, early studies reporting outcomes in COVID-19 identified that elevated clinical inflammatory markers were prognostic of disease severity and mortality. In addition higher serum ferritin levels were reported in patients with thrombotic complications than in the others and the hyperferritinemia correlates well with the severity of the COVID-19, and the serum ferritin levels increase during the aggravation of the infection. The antiviral properties of lactoferrin are now known.

However, the paper has several flaws and I do not consider it suitable for publication on Nutrients in its current form.

The introduction is necessary and here, it is necessary to better clarify the aim of this manuscript. Is it a narrative review? Why did you choose to analyze these substances?

In addition, I suggest that we delete paragraph “Iron Homeostasis and Lung Injury”,I think it’s confusing for the reader. The topic is the evaluation of Iron Homeostasis, Anemia of Inflammation, Lactoferrin, and  viral infection with a specific topic regarding 2 SARS-CoV-2 Infection, is that right?

In the paragraph “Antiviral Activity of Lactoferrin” I recommend mentioning a recent review about this topic “Sinopoli, A, Isonne, C, Santoro, MM, Baccolini, V. The effects of orally administered lactoferrin in the prevention and management of viral infections: a systematic review. Rev Med Virol. 2022; 32( 1):e2261. https://doi.org/10.1002/rmv.2261”.

Author Response

Reviewer 3:

              We thank the Reviewer for their insightful review, comments, and suggestions for revision of our manuscript. Based on this review and others, we have substantially revised the manuscript and hope Reviewer 3 finds it much improved. Specifically, we hope that by reorienting the focus and narrowing the scope, this revised version is more readable and informative. Specific to Reviewer 3 recommendations, we have added an introduction that we hope brings more clarity to the aim of the manuscript.

              All three reviewers felt that the paragraphs on “iron homeostasis” and “iron homeostasis and lung injury” needed substantial revision. Based on feedback from all three reviewers, we have removed the paragraph “iron homeostasis and lung injury”, as the reviewers felt the paragraph was out of scope for the manuscript and only added to their confusion. We thank Reviewer 3 for this suggestion.

              In the paragraph “Antiviral Activity of Lactoferrin”, we have incorporated the suggested review by Sinopoli et al. In addition, we have added the review and references from the review to Table 1. We very much appreciate Reviewer 3 recommending this citation, as the information in this systemic review supports and validates our perspective that lactoferrin has a role as a therapeutic adjuvant during viral infection.

Again, we thank Reviewer 3 for their comprehensive review and believe this revised manuscript is improved by incorporation of their suggestions and comments.

Round 2

Reviewer 1 Report

The manuscript has been improved and it can be accepted in the present form

Author Response

We thank Reviewer 1 for their positive review and hope they find this revised version even more improved. Again, we thank Reviewer 1 for their insightful review of our manuscript and think the revised version was improved by they input and expertise.

Reviewer 2 Report

The authors have followed many of my suggestions and the manuscript is much improved and easier to read.

Line 33. I am not sure enterocytes can be described as “iron overloaded”. As iron absorption is decreased by inflammation iron remains in the enterocytes but is excreted after a few days as the enterocytes are sloughed off and pass into the stool.

Line 45 “hephaestin” should be correctly spelt.

Line 63: only free iron in the body can induce free radicals. Is there any evidence of free iron being generated during viral infections?

Figs 1 and 2. The figures are artistically very good and very colourful however the reader needs a lot of imagination to follow the aims of the authors. In general, all the abbreviations should be clearly defined. I am not convinced that the figures help or add to the understanding of the text. I am not convinced of their usefulness.

Line 262 ferroportin

Line 359. Please be more specific as to the method of “feeding” or administrating the lactoferrin. If it is administered via the intestine how is it absorbed. If it is digested, will it still be active? Please discuss this issue.

Author Response

Response to Reviewer 2:

We thank Reviewer 2 for their comprehensive and insightful review of our revised manuscript. We appreciate the expertise and critical insight provided by Reviewer 2 and hope they find this second revision an improvement upon the first revision. We believe that the constructive criticism and thoughtful insight from Reviewer 2 has improved the value and impact of this manuscript.

Specific concerns addressed in this revision include:

  • We agree that there is not strong evidence to differentiate between enterocyte iron overload and iron homeostasis impacts from enterocyte iron uptake and release as enterocytes are sloughed off into the stool during inflammation. We have edited the manuscript to refer only to macrophage iron overload, as this is well supported by the literature.
  • Thank you to Reviewer 2; we have corrected the spelling of hephaestin.
  • While the impact of circulating or intracellular free iron during viral infection is still an area of active investigation, in chronic hepatitis there is evidence that the viral infection results in higher levels of circulating free iron. We have included this reference.
  • We appreciate that Reviewer 2 complements the artistic quality of Figures 1 and 2. We are apologetic that Reviewer 2 does not find the figures as helpful or adding to the text of the manuscript; however, given the feedback from the other Reviewers, we have chosen to retain the figures as presented. In response to Reviewer 2 critique, we have added the missing abbreviations to Figure 2, as requested, and hope this improves the figure.
  • We thank Reviewer 2 for catching our misspelling of ferroportin and have corrected this error.
  • We found Reviewer 2 reference to Line 359 a bit confusing and apologize; however, we have added a paragraph at the end of the Line 359 paragraph further exploring the potential utility of administering oral Lf. While we did not find good information on activity of oral Lf systemically, we did cite a recent study that looked at the efficacy of digested Lf against SARS-CoV-2 infection in vitro. We hope this brief discussion will provided needed insight into both the need for more research into systemic impact of oral Lf and the specific utility of oral Lf as a therapeutic adjuvant for COVID-19.

Reviewer 3 Report

The improvements are sufficient.

Author Response

We thank Reviewer 3 for their positive review of our revised manuscript and hope they find this manuscript a valuable contribution to the field. We appreciate the insight and expertise provided by Reviewer 3 and feel the manuscript was improved by their input.